# Two-Stage Atomic Decomposition of Multichannel EEG and the Previously Undetectable Sleep Spindles

**DOI:** 10.3390/s24030842

**Published:** 2024-01-28

**Authors:** Piotr Durka, Marian Dovgialo, Anna Duszyk-Bogorodzka, Piotr Biegański

**Affiliations:** 1Faculty of Physics, University of Warsaw, 02-093 Warsaw, Poland; marian.dovgialo@fuw.edu.pl (M.D.); pbieganski@fuw.edu.pl (P.B.); 2Behavioural Neuroscience Lab, Institute of Psychology, SWPS University, 03-815 Warsaw, Poland; aduszyk1@swps.edu.pl

**Keywords:** EEG, time-frequency, EEG inverse solution, atomic decomposition, sleep spindles

## Abstract

We propose a two-step procedure for atomic decomposition of multichannel EEGs, based upon multivariate matching pursuit and dipolar inverse solution, from which atoms representing relevant EEG structures are selected according to prior knowledge. We detect sleep spindles in 147 polysomnographic recordings from the Montreal Archive of Sleep Studies. Detection is compared with human scorers and two state-of-the-art algorithms, which find only about a third of the structures conforming to the definition of sleep spindles and detected by the proposed method. We provide arguments supporting the thesis that the previously undetectable sleep spindles share the same properties as those marked by human experts and previously applied methods, and were previously omitted only because of unfavorable local signal-to-noise ratios, obscuring their visibility to both human experts and algorithms replicating their markings. All detected EEG structures are automatically parametrized by their time and frequency centers, width duration, phase, and spatial location of an equivalent dipolar source within the brain. It allowed us, for the first time, to estimate the spatial gradient of sleep spindles frequencies, which not only confirmed quantitatively the well-known prevalence of higher frequencies in posterior regions, but also revealed a significant gradient in the sagittal plane. The software used in this study is freely available.

## 1. Introduction

The gold standard and common benchmark for the interpretation of electroencephalographic (EEG) recordings is based upon visual inspection of EEG traces, presented in the time domain. This approach encompasses decades of worldwide clinical and research experience, providing an invaluable knowledge base of neurophysiological and behavioral correlates. This knowledge is mostly phenomenological, and relates to the *appearance* of EEG structures—“graphoelements” [1]. Time domain analysis was supplemented by the automatic calculation of the frequency content as early as 1932 [2], which led to the recognition of the major EEG rhythms. However, power spectrum reflects the average properties of an epoch, losing the information about its time dynamics. This, in turn, led to the application of time-frequency methods. Finally, the availability of multichannel EEG recordings (Figure 1, upper panel) revealed the spatial distribution of scalp potentials. However, to determine the anatomical and functional brain areas generating the observed structures, we must solve the so-called EEG inverse problem which, per se, is ill-posed; that is, the solution is non-unique and does not change continuously with input data. Its regularization can be based on several different assumptions, like the minimum norm or maximal smoothness of the solution, leading to a variety of algorithms for EEG inverse solutions. Another, less widely discussed problem relates to the fact that EEGs contain contributions from all the simultaneously active brain generators, so at any moment, the generators contributing to the recorded signal can be spread across the whole brain volume. Their number is unknown a priori, and significantly exceeds the number of recording electrodes, making the problem heavily underdetermined. Therefore, the first step of a reasonable solution should include extraction from the multichannel EEG time series of relevant activities, possibly related to as few neural generators as possible [3].

Several attempts to describe multichannel EEG simultaneously in time, frequency, and space have been proposed and exemplified on simulated and small exemplary datasets; e.g., in 1996, A.B. Geva proposed multivariate decomposition of evoked potentials in a wavelet frame as an input to dipolar inverse solutions, which “converged on physiologically and anatomically reasonable sources” [4]. In 2001, Koenig et al. [5] proposed a “topographic time-frequency decomposition” enhancing the EEG microstates [6] by their time-frequency signatures. In 2016, Kortas et al. [7] added a spatial dimension to the time-frequency dictionary obtained by thresholding a common part of separate wavelet decompositions of EEG channels. Finally, in 2017, Kordowski et al. [8] optimized a matching pursuit search in a product of spatial distributions of ∼$10^{5}$ MEG fields of cortical dipoles with a discrete dictionary of ∼$10^{5}$ time domain chirplets.

The proposed approach is built on adaptive time-frequency decomposition of EEG graphoelements, which was introduced in 1995 [9] and, since then, confirmed in several studies (c.f. [10] and references therein), and in our previous attempt to combine it with distributed inverse solutions [11]. It offers a high-resolution sparse atomic time-frequency-space decomposition of multichannel EEG recordings, based on multivariate matching pursuit (MMP, [10]) and dipolar inverse solutions, compatible with the traditional, visual EEG analysis. It consists of two major stages:Multichannel EEG recordings are first decomposed of the MMP algorithm into waveforms (called atoms) of well-defined time span and position, frequency, phase, and a vector of amplitudes in recorded derivations, which determines the spatial distribution on the scalp.Assuming that an oscillation of constant frequency and phase reflects the activity of at most one neural generator, we fit a single equivalent current dipole model to the atom’s amplitudes.

From this pool of time-frequency atoms with estimated spatial locations of corresponding brain generators, we select those conforming to the properties of relevant structures, using the knowledge of neurophysiology and visual EEG analysis.

As an example application, we present automatic detection and parametrization of sleep spindles, based directly upon their definition from “The AASM manual for the scoring of sleep and associated events” [12]: “*A train of distinct sinusoidal waves with a frequency of 11–16 Hz (most commonly 12–14 Hz) with a duration ≥0.5 s, (…)*”. We compare the performance of the proposed method to the gold standard of visual detection and recently published algorithms.

The presented results suggest that even 2/3 of structures conforming to the classical definitions were previously undetected by both human scorers and algorithms. As a demonstration of the potential of the fully automatic and detailed parameterization of all the spindles in many overnight recordings, we compute the spatial gradient of spindles frequencies, which confirms the well-known prevalence of faster spindles in posterior regions and reveals a previously unobserved gradient in the sagittal plane.

## 2. Materials and Methods

### 2.1. Multivariate Matching Pursuit (MMP) Decomposition

In this study, we decomposed 19-channel EEG recordings resampled to 128 Hz (Section 2.3). Denoting a 20 s epoch as x (in this case a 2560 × 19 matrix), and the 2560 points in the *i*th channel (i=1…19) as $x^{i}$, the variant of MMP decomposition applied in this study can be described by the following recursive equations:(1)$$\left\{\begin{array}{c} R^{0}x=x\\ R^{n}x^{i}=\langle R^{n}x^{i},g_{\gamma_{n}}\rangle g_{\gamma_{n}}+R^{n+1}x^{i}\\ g_{\gamma_{n}}=\arg\max_{g_{\gamma }}\sum_{i=1}^{19}{\left|\langle R^{n}x^{i},g_{\gamma }\rangle \right|}^{2} \end{array}\right.$$

This recursion was stopped after 200 iterations, yielding an adaptive approximation of the original signal in terms of a linear sum of chosen functions $g_{\gamma_{n}}$:(2)$$x^{i}\approx \sum_{n=0}^{199}\langle R^{n}x^{i},g_{\gamma_{n}}\rangle g_{\gamma_{n}}$$

As the basic decomposition blocks (time-frequency atoms) $g_{\gamma }$ we chose sinusoidal oscillations modulated by Gaussian envelopes (Gabor functions):(3)$$g_{\gamma }\left(t\right)=K\left(\gamma \right)e^{-\pi {\left(\frac{t-u}{s}\right)}^{2}}\cos\left(2\pi f(t-u)+\phi \right),$$
where K(γ) is such that $\left|\right|g_{\gamma }\left|\right|=1$, and the vector of parameters γ={u,f,s,ϕ} denotes time and frequency centers, width and phase. $\langle R^{n}x^{i},g_{\gamma_{n}}\rangle$ are the weights of the selected function $g_{\gamma_{n}}$ in respective derivations, which determine the scalp topographies of corresponding structures, used for dipole fitting described in the next section.

A potentially cross-term free time-frequency distribution of energy density of channel *i* can be computed from this decomposition as
(4)$$Ex^{i}=\sum_{n=0}^{199}{\left|\langle R^{n}x^{i},g_{\gamma_{n}}\rangle \right|}^{2}Wg_{\gamma_{n}}$$
where $Wg_{\gamma_{n}}$ is the Wigner distribution of $g_{\gamma_{n}}$ [13].

In this study, we used a recent MMP implementation [14,15] (available from https://github.com/develancer/empi/, accessed on 8 February 2022) with global dictionary optimization, 200 iterations, energy error = 0.05, Gabor width 0.1–10 s, and maximum frequency 45 Hz.

### 2.2. Dipole Fit

Equation (Equation 1) approximates multivariate EEG as a linear sum of time-frequency atoms $g_{\gamma_{n}}$, each of them accompanied by the vector of amplitudes in respective derivations ${\left\{\langle R^{n}x^{i},g_{\gamma_{n}}\rangle \right\}}_{i=1\ldots 19}$. This vector represents the scalp topography of $g_{\gamma_{n}}$. If such a structure originates from a generator within the brain, these topographies result from Maxwell’s equations, with boundary conditions representing the electrical properties of the tissue between the source and recording electrodes. Using this assumption, we can estimate the spatial coordinates of the source by solving the EEG inverse problem.

To fit dipole sources to the scalp topographies of each time-frequency atom, we used the mne.fit_dipole function from the MNE Python library ([16], https://mne.tools/, accessed on 10 August 2022 ). To remove dependence on reference electrode placement, the amplitudes were first re-referenced to the average reference.

Dipole fit was based on a boundary element method (BEM) model derived from segmentation of *fsaverage* MRI atlas, originally provided in FreeSurfer software [17], and generic 10–20 electrode positions in MNI coordinates. Finally, we used the *fsaverage* MRI atlas parcellation to calculate the dipole’s distance to the closest cortical voxel. In this way, the original set of parameters from Equation (Equation 3), described each structure from the linear MMP expansion (Equation 2); that is: time center *u*, frequency center *f*, time width *s*, phase ϕ, and  scalp distribution of amplitudes ${\left\{\langle R^{n}x^{i},g_{\gamma_{n}}\rangle \right\}}_{i=1\ldots 19}$, was extended by: the dipole’s location (MNI coordinates), the dipole’s direction angle, the goodness of fit (percentage of scalp energy explained by the dipole), the dipole’s current density, and the dipole’s distance from the cortex.

In the following, a structure denoted by the above sets of parameters and properties is called a *dipole atom*, and the above procedure is called *MP-dip*.

### 2.3. Experimental Data

The Montreal Archive of Sleep Studies (MASS, [18], http://ceams-carsm.ca/en/mass/, accessed on 20 April 2022) is an open access database of fully anonymized laboratory PSG recordings of healthy adults, with sleep stages marked by experts, collected in the period between 2001 and 2013 at three different laboratories of the Center for Advanced Research in Sleep Medicine (CARSM) based in Montreal, Canada. A total of 160 of these recordings (cohorts S1, S2, S3, and S5) contained all the derivations from the 10–20 system, necessary for the estimation of dipole sources. The usage of these data were approved by the Rector’s Committee for the Ethics of Research Involving Human Participants at the University of Warsaw (Application No. 104/2021).

Massive Online Data Annotation (MODA) study ([19], https://github.com/klacourse/MODA_GC, accessed on 20 April 2022) provides annotations of the MASS dataset, reflecting the consensus of a group of experts, researchers, and non-researchers marking sleep spindles in selected epochs of artifact-free N2 sleep stages. Annotations were based solely on the C3 channel referenced to the left ear. For comparison with detections of discussed algorithms, we used the consensus of the group of experts. The intersection of MASS recordings with a full 10–20 electrode set and MODA annotations consisted of 147 recordings.

In total, 19 EEG channels from the above-described recordings were resampled to 128 Hz and bandpass filtered from 0.5 to 30 Hz using two-sided infinite impulse response filtering, fourth order Butterworth filter. These multivariate time series were split into 20 s epochs and subjected to MMP decomposition and subsequent dipole fitting, as described in the preceding Section 2.1 and Section 2.2.

From these decompositions, sleep spindles were selected as conforming to the criteria from Table 1, and chosen to reflect the available prior knowledge [12]. Figure 2 presents a flowchart of this procedure.

### 2.4. Decomposition of a Sample EEG Epoch

The procedure described in the previous section is exemplified in Figure 1 on a sample 20 s epoch of 19 channels of EEG from the subject 01-05-0015 from MASS. The gray rectangle in the upper panel (approx. 2.5–5 s) marks a single sleep spindle according to the consensus of experts from MODA. It comprises three structures detected as spindles by *MP-dip*, marked A, B, and C on the distribution of energy density (Equation (Equation 4)) of the C3–Linked Ears derivation, shown below. Green arrows point from these structures toward the inserts showing scalp distributions of their activities, estimated by MMP (Equation (Equation 2)) and extrapolated. Three-dimensional representations of equivalent dipoles, fitted to these distributions, are shown below.

The structure was marked as one single spindle by an expert, ignoring the underlying microstructure of the three potentially different phenomena, contributing to the shape perceived by the expert as one spindle. Structures A and B have similar frequencies and spatial locations of equivalent dipolar sources, so it can be argued that we may deal, e.g., with one chirp-like structure of frequency changing with time [20]. However, structure C is apparently distant from these two in both the frequency and the spatial location of the source.

## 3. Results

### 3.1. Comparison of Detection of Sleep Spindles with Experts and Automatic Detectors

As the reference for verification of the detection of sleep spindles, we used annotations reflecting the expert group consensus based upon visual inspection of the C3 derivation from N2 NREM sleep, from the Massive Online Data Annotation (MODA, [19]). Spindles marked by the algorithms, as well as annotations from MODA, were filtered to satisfy the minimum textbook spindle length of 0.5 s; that is, shorter annotations were rejected. We also compared the performance of detection with two recently published algorithms, representing the state-of-the-art in automatic detection of sleep spindles: YASA—Yet Another Spindle Algorithm [21], and SUMO—Sleep U-Net trained on MODA [22].

All these automatic detections were compared to the MODA markings, treated provisionally as the “ground truth”; that is, automatic detections—by SUMO, YASA or *MP-dip*—were treated as true positives (TP) if found in a position marked also by MODA experts; all other automatic detections contributed to the “formally FP” (false positive) cases.

However, to compare different markings, we must take into account one crucial parameter governing the basic rules of scoring two markings as “concordant”. Namely, algorithms—as well as human scorers—provide markings in terms of estimated time spans of sleep spindles; that is, start and end times. Annotations of the same spindle, produced by scorers and algorithms, may overlap more or less. The least restrictive scoring counts two annotations as related to the same spindle if they overlap by at least one point. The performance of the investigated algorithms depending on this overlap is presented in the right panel of Figure 3.

Another issue relates to the fact that human experts—and hence also the algorithms replicating their decisions—tend to miss the structures buried locally in noise. The concordance of algorithmic markings with human detections can be artificially improved by the rejection of structures with a low signal-to-noise ratio (SNR), as presented in Figure 3. For each selected $g_{\gamma }$, SNR was computed as
(5)$$\mathrm{SNR}=\;\frac{\left|\right|\langle R^{n}x^{1},g_{\gamma_{n}}\rangle {\left|\right|}^{2}}{\left|\right|{\mathrm{EEG}}_{u\pm s}{\left|\right|}^{2}}$$
where ${\mathrm{EEG}}_{u\pm s}$ is the preprocessed epoch of the background EEG around $g_{\gamma }$, from u−s to u+s. “Background” is taken explicitly as not containing the signal (spindle), hence $\langle R^{n}x^{1},g_{\gamma_{n}}\rangle g_{\gamma_{n}}$ is subtracted from the signal before computing $\left|\right|{\mathrm{EEG}}_{u\pm s}{\left|\right|}^{2}$.

The usage of this parameter requires special attention. Formally, it is not present in the definitions of sleep spindles. It depends on the energy of other brain activities and noise, occurring at the same time as the spindle. Low SNR obscures the visibility of spindles to a human scorer but, as such, does not reflect any intrinsic property of spindle or other structure. Although the aim of the proposed approach is beyond the blind replication of human scorer’s decisions, for subsequent comparison with other methods, we also present results for “clearly visible” structures with SNR above −7 dB, giving maximum concordance as shown in the left panel of Figure 3.

Unsurprisingly, the best concordance with human scorers (F1 = 0.79) was achieved by the SUMO algorithm—a deep neural network model optimized for reproducing the experts’ markings from the same MASS database with MODA annotations. The second-best result (F1 = 0.67) was obtained by the YASA toolbox, where detection of sleep spindles is based upon modified A7 algorithm from [23], which uses thresholding of EEG signal sigma spectrum features with parameters optimized for concordance with experts. The plain *MP-dip*-based algorithm achieved F1 = 0.47, inferior to both of the above. However, this algorithm was entirely based upon a priori textbook knowledge, not optimized for concordance with any dataset or scorer. We can easily boost its performance by rejecting detections of structures “hidden in noise”; that is, structures with a local signal to noise ratio lower than the optimal value from the left panel in Figure 3. Adding such a filter, simulating the main weakness of the other algorithms, gives a performance close to YASA (green dots in the right panel of Figure 3, F1 =0.63).

Plain *MP-dip* detected, on average, three times more spindles than human experts or the other algorithms. This is mostly due to the inclusion of structures hidden in the noise, and hence elusive to both the human experts as well as algorithms trained to blindly replicate their decisions. To investigate the properties of these formally “false positive detections”, in Figure 4, we compared normalized histograms of all the TP and FP spindles detected by plain *MP-dip*. In the lower panel of Figure 4 we present the *p*-values for the hypotheses of TP and FP cases coming from the same probability distribution in each of the analyzed recordings separately. For the time-frequency and spatial parameters, the hypothesis can be rejected at p=5% in about 5% of the recordings, which confirms the fact that both the “TP” and “FP” spindles share the same properties. The only difference between the “false positives” and the spindles detected by human scorers lies in their visibility, reflected in the histograms of local signal to noise ratios and, indirectly, in the amplitudes.

To evaluate these effects quantitatively, we tested the hypotheses of equal distributions of these parameters in each of the 144 (out of 147) recordings, containing large enough numbers of spindles marked by experts to reliably compute statistics. The lower panel of Figure 4, presents *p*-values for the hypotheses of TP and FP—in each recording separately—coming from different distributions. We observe that only for SNR and amplitudes, large fractions of subjects (represented by dots) fall below the 5% line (89 and 96 out of 144). For the time-frequency-space parameters, the fractions are 5/144, 32/144, 29/144, 15/144, and 21/144, respectively (which is expected for a statistical test at a *p*-level of 5%), for phase, frequency and so on from the left in the bottom panel of Figure 4. This suggests that at least a large proportion of sleep spindles, treated as FP in relation to the human detections, may indeed represent the same phenomenon as TPs, and their omission stems only from an unfavorable local SNR.

### 3.2. Spatial Gradient of Frequencies of Sleep Spindles

The above results encourage us to further investigate the properties of detected sleep spindles. Let us take a generally known fact, that faster (higher frequency) spindles appear in posterior regions [24,25]. Within the proposed framework, we can test a more general and detailed hypothesis of a change (gradient) of spindle frequencies across the spatial coordinates.

The spatial locations of dipole atoms representing sleep spindles and their corresponding frequencies can be treated as a sparsely sampled scalar field. We will estimate the mean gradient of this field by fitting a linear function:(6)f(x,y,z)=jx+ky+lz+c
where *f* is the frequency in Hz; *x*, *y*, and *z* are dipole coordinates in MNI coordinate frame in millimeters; and *c* is a constant. Using the least squares fit of dipole locations and respective frequencies, parameters j,k,l are estimated and treated as a three-dimensional vector, which points towards the average gradient of this scalar field in *x*, *y*, and *z* coordinates with units of hertz per millimeter. This vector is normalized to unit length, and the dot product is calculated with each dipole source location, to produce a new positional coordinate along the gradient direction. Afterwards, the Spearman correlation coefficient is computed for a correlation between spindle dipole position and frequency and a *p*-value of a null hypothesis of two non-correlated systems achieving this level of correlation.

As presented in the upper left panel of Figure 5, the direction of the gradient in the axial plane confirms the known trend of the prevalence of faster spindles (green dots) in the posterior regions. However, projections on the two other planes also reveal consistent differences in the spatial distribution of frequencies. The middle row shows the gradient vectors calculated for each of the subjects as blue arrows, with their averages in red. Their directions are consistent across the subjects, indicating the overall trend of increasing spindles frequencies not only toward the back of the head but also upwards.

Owing to the detailed estimation of positions and frequencies of each single sleep spindle, we may estimate the correlations between their frequencies and positions along the three major axes (X, Y, Z) and along the direction of the gradient. The lower left panel of Figure 5 presents boxplots of these correlations computed separately for each subject, while the boxplots in the lower right panel indicate *p*-values of these correlations. Except for the X direction (left-right differences between hemispheres), very few subjects reveal *p*-values exceeding 0.05 (dashed horizontal line).

### 3.3. Simulations and Accuracy

Section 3.1 compared the performance of the proposed approach to *detections* of human experts, treated provisionally as a ground truth. However, even such an approximate “ground truth” is not available for the *parameters* of electroencephalographic sleep spindles. We assess the accuracy of their estimation on simulated signals, constructed as follows.

Background EEG was generated from a multivariate autoregressive model (MVAR) of order 14 fitted to a fragment of N2 stage from one of the analyzed recordings. To this signal, we linearly added Gabor functions representing sleep spindles with fixed frequencies and time spans (ω and *s* from Equation (Equation 3)) at 14.5 Hz and 3 s, respectively. Twenty dipolar sources were distributed in the MNI coordinates along a line in the *y*–*z* plane, while the values for the *x* coordinates were taken from a normal distribution with zero mean and standard deviation $\frac{1}{150}$ m. These parameters were chosen to represent both deep and shallow sources, with five random orientations each. Fifteen amplitudes of these dipoles were linearly distributed between 75 nAm and 150 nAm.

Values of the scalp amplitudes of spindles were taken from a forward solution computed for these dipoles using the “average brain” model (*fsaverage* MRI atlas, originally provided in FreeSurfer software [17]). In such a way, 1500 spindles were placed within 225 epochs of simulated 20 s 20-channel EEG. For each of the simulated spindles, SNR was computed according to Equation (Equation 5).

These signals were subjected to the same analysis as the EEG. To simulate the lack of the MRI scans corresponding to the analyzed EEG recordings, we also computed the inverse solutions (Section 2.2) for a different brain model, taken from the sample dataset included in the MNE software [16]. In such a way, equivalent dipoles were computed in two different ways: (1) using the same model (lead field) as in the forward simulation, in the following labeled “matching MRI”, and (2) using the lead field from a different MRI (“other MRI”). As a result, 1353 of 1500 simulated structures were found to conform to the criteria from Table 1 for matching MRI, while for the “other MRI”, only 1175 spindles were detected. Their SNRs lay mostly between −23 and 5, hence the ranges of the axes in Figure 6 and Figure 7. However, values in Table 2 were computed for all the detected structures.

Errors of the spatial locations of dipoles were computed as the Euclidean distances between the locations of the simulated and fitted dipoles. For the remaining parameters, errors were computed as the absolute differences between the simulated and estimated values (amplitude on the Fp1 derivation).

## 4. Discussion and Conclusions

Results presented in the previous section—Table 2 and Figure 6—confirm the amazing accuracy of the MMP in retrieving the time-frequency parameters of the simulated sleep spindles even for very large SNRs, which is partly due to the fact that Gabor functions are included in the dictionary used in the decomposition. However, errors of the spatial location of generators (Figure 7) are still relatively large. While the superior performance of the MMP over other methods of data preprocessing for the EEG inverse solutions was shown in [3], these errors seem to reflect the inherent problems of spatial estimation of electroencephalographic sources [26].

The proposed approach detects on average about three times more spindles than both human experts and the algorithms designed to blindly replicate their markings. This is a result of the unparalleled accuracy of recovering the original parameters of spindles by the MMP algorithm, even in very small signal-to-noise ratios, which was shown on the simulated signals in Section 3.3. These previously undetected “false positive” spindles share a priori all the measurable properties of “true positives”, including frequency, duration, and amplitude: as presented in Section 3.1; parameters of the “false positive” spindles come from the same distributions as those detected by human experts at 5% significance level, except for the SNR and amplitude—“invisible” spindles tend to occur in lower SNRs, correlated also with lower amplitudes.

Presented results suggest that all the spindles detected within the proposed approach reflect the same physiological phenomena, and, consequently, that the previously applied approaches missed up to 2/3 of the sleep spindles, detectable in EEG according to the standard criteria [12].

Finally, the detailed parametrization of sleep spindles in time, frequency, and space coordinates, possible within the proposed approach, allowed us to compute a spatial gradient of their frequencies, presented in Section 3.2. Apart from the confirmation of the well-known prevalence of slower spindles in frontal regions, we observe for the first time significant gradients of frequencies in the sagittal plane, as presented in Figure 5.

Such a detailed, quantitative, and automatic analysis of thousands of sleep spindles and their time-frequency and spatial parameters is, to our knowledge, presented for the first time, and was not possible within the previously known approaches. It exemplifies the new possibilities in EEG/MEG analysis opened by the presented method, designed to unify the indispensable knowledge base of behavioral and clinical correlates of EEG, collected in decades of visual EEG analysis, with the recent advances in neurosciences. In this study, the former is represented by the textbook definition of sleep spindles, which was implemented directly and explicitly. Efficient and robust implementation of the proposed ideas was made possible by the recent advances in mathematics, informatics, physics, and neurophysiology.

## Figures and Tables

**Figure 1 sensors-24-00842-f001:**
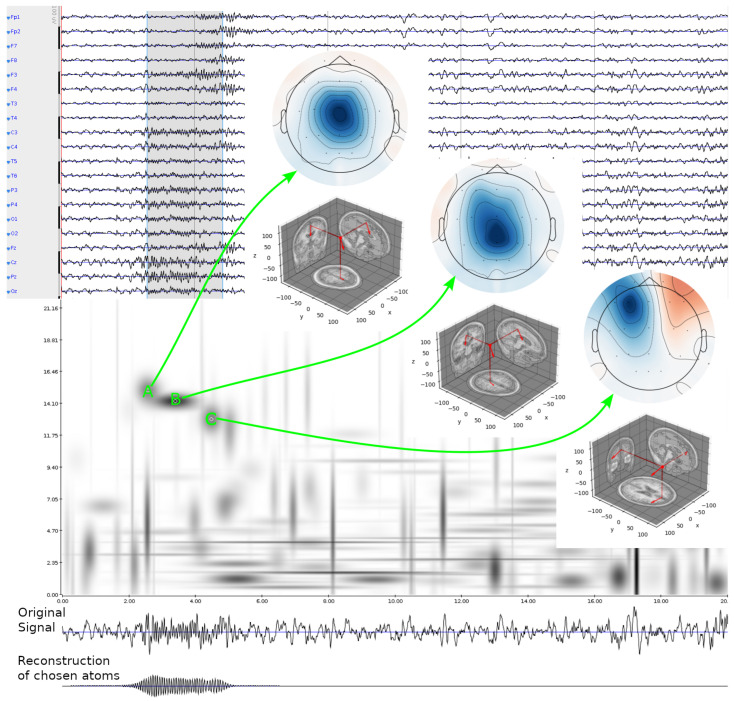
Upper background panel: 20 s of sleep EEG, 19 derivations, subject 01-05-0015 from MASS; the gray rectangle reflects MODA marking of a sleep spindle. Lower background: time-frequency energy density (Equation (Equation 4)) of the C3 derivation. Front: green arrows point from the time-frequency blobs, representing the structures detected as spindles by *MP-dip*, towards distributions of their amplitudes on the scalp; below each distribution, the estimated dipolar source is presented in 3-D. Lowest trace: sum of the Gabor functions (Equation (Equation 3)) corresponding to A, B, and C.

**Figure 2 sensors-24-00842-f002:**
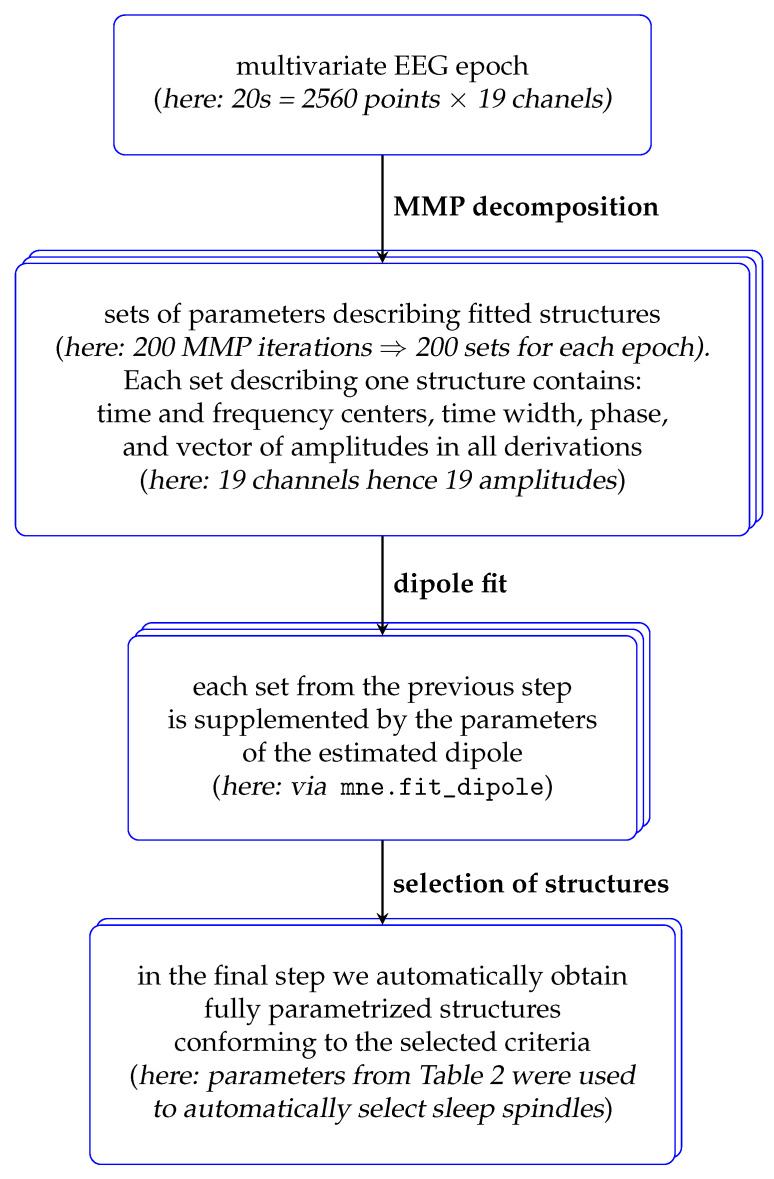
A general flowchart of the proposed method; “*here*:” refers to the detection of sleep spindles as described in this study.

**Figure 3 sensors-24-00842-f003:**
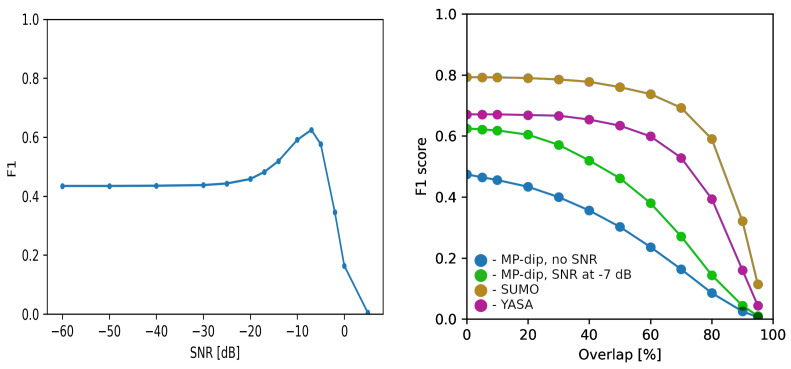
(**Left panel**) concordance of spindle detection with human experts, measured by the F1 score, depending on the structure’s minimum signal-to-noise ratio (Equation (Equation 5)) of a structure accepted as a spindle. (**Right panel**) F1 score (vertical) vs. minimal overlap (horizontal) of spindles detected by the experts and algorithms.

**Figure 4 sensors-24-00842-f004:**
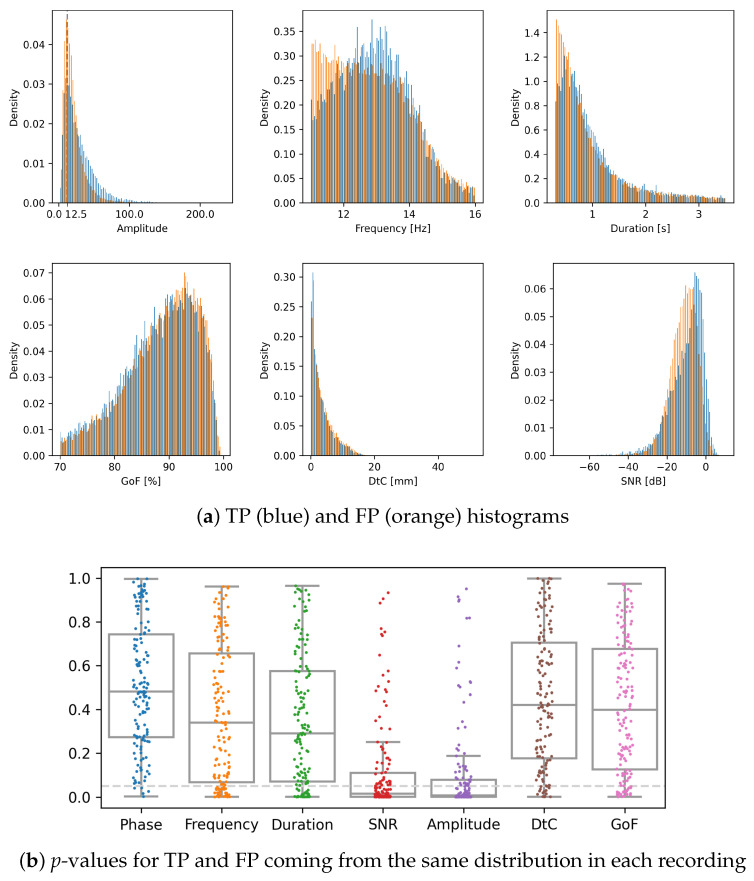
Upper panel (**a**) normalized histograms of True Positive (blue, 6508 cases) and False Positive (orange, 14052 cases) spindles detected by MP-dip algorithm without SNR threshold. Lower panel (**b**) *p*-values of two-sample Kolmogorov–Smirnov test for the hypothesis of TP and FP spindles coming from the same distribution, with each dot representing a parameter/recording pair. DtC—distance to cortex, GoF—goodness of fit of the dipole. The horizontal gray line is drawn at p=0.05.

**Figure 5 sensors-24-00842-f005:**
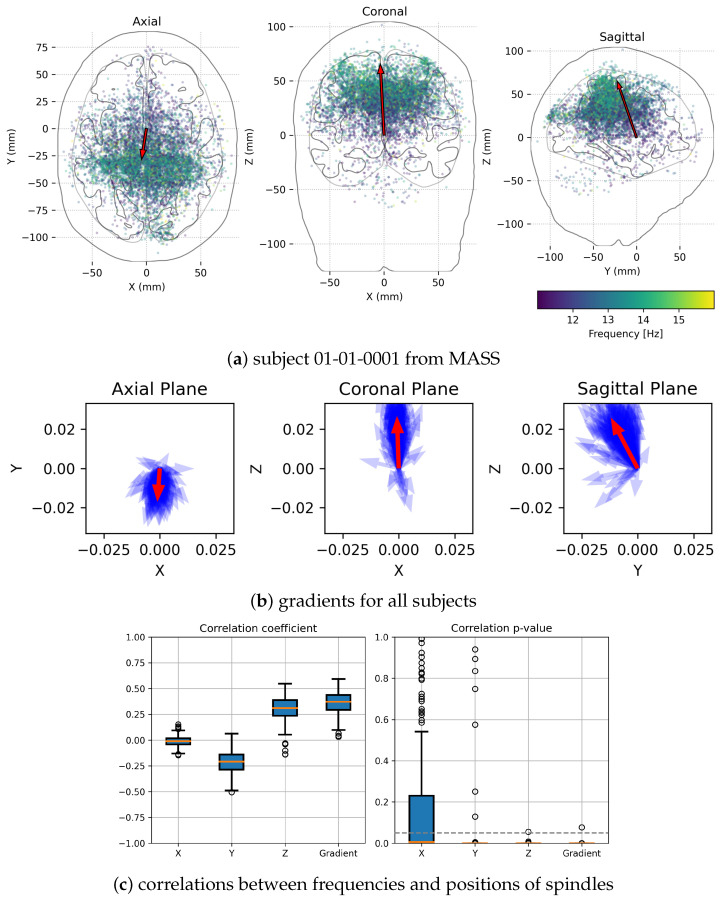
From the top: (**a**) sleep spindles detected in stage N2 of a single recording (01-01-0001); dots represent single spindles, colors reflect their frequencies. Superimposed on the contours of sections of the cortex (without cerebellum, etc., hence some dots seem to fall outside the brain). Red arrows indicate frequency gradients. Middle (**b**) gradients ([Hz/mm], Equation (Equation 6)) for each of the subjects (blue) and their averages (red). Bottom (**c**) boxplots of correlation coefficients between frequencies and positions of spindles, and their *p*-values, computed for all subjects along 3 major axes, and along the gradient direction. Orange line shows the median value, blue boxes—the first and third quartiles.

**Figure 6 sensors-24-00842-f006:**
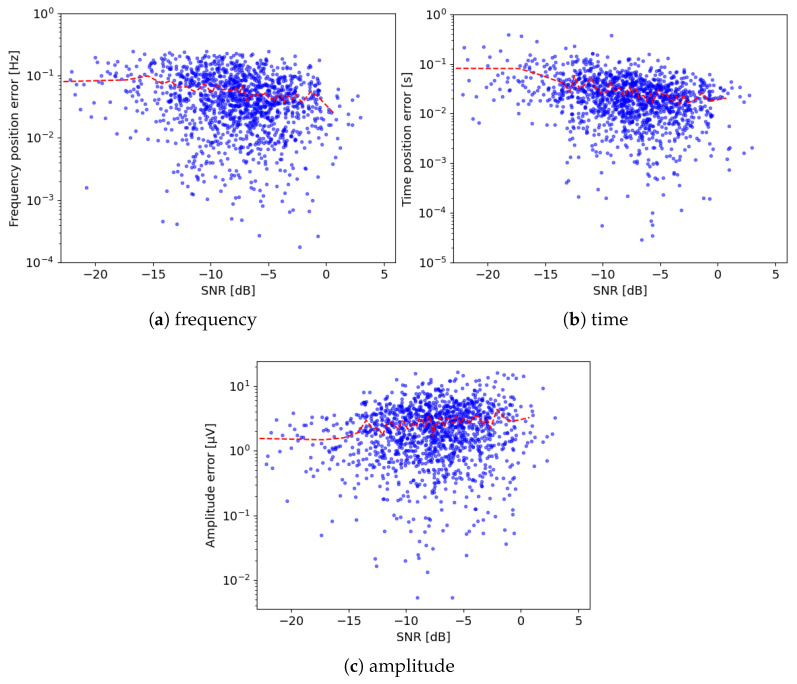
Errors of estimated parameters: frequencies (**a**), times (**b**), and amplitudes (**c**) on Fp1 derivation of simulated sleep spindles. Red dotted lines represent smoothed means for given SNR ranges, means of these means are given in Table 2.

**Figure 7 sensors-24-00842-f007:**
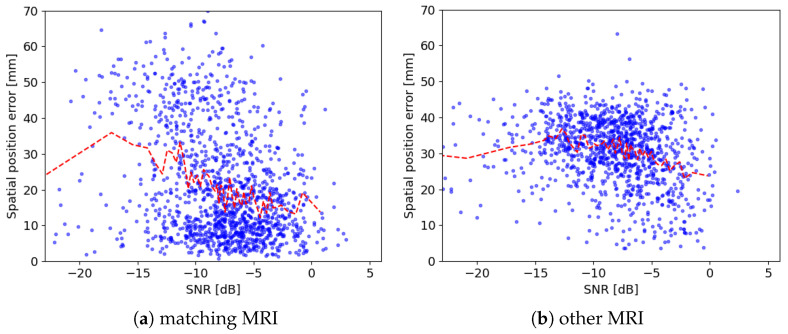
Errors of the spatial location of simulated generators of sleep spindles for the cases when the inverse dipolar solution was computed using the same lead field as used in simulations (corresponding to using “matching MRI”, left panel (**a**)) or a different lead field (“other MRI”, right panel (**b**)). Red lines represent the means, also presented in Table 2.

**Table 1 sensors-24-00842-t001:** Ranges of parameters used for selecting sleep spindles from the *dipole atoms* fitted by *MP-dip*. The number of oscillation periods is counted within the width of a structure (*s* from Equation (Equation 3)).

Parameter	Value Range
highest amplitude (across derivations)	12.5–150 μV
frequency	11–16 Hz
duration	0.5–3.5 s
number of oscillation periods	3–*∞*
dipole goodness of fit (GoF)	60–100%
dipole distance to nearest cortical voxel (DtC)	0–20 mm

**Table 2 sensors-24-00842-t002:** Mean and median errors of parameters estimated for simulated signals, averaged for SNR between −40 and 5 dB.

	Mean Error ± STD	Median Error
Frequency [Hz]	0.056±0.049	0.041
Time [s]	0.031±0.035	0.022
Amplitude Fp1 [µV]	2.71±2.42	2.10
matching MRI		
Location [mm]	20.85±15.26	16.27
other MRI		
Location [mm]	30.78±9.46	31.98

## Data Availability

Source code for the proposed detection algorithm, including scripts for the analyses presented in this paper, is available at https://gitlab.com/fuw_software/mp-dip (accessed on 1 November 2023) under GPLv3 license. We used the implementation of MMP available from https://github.com/develancer/empi (accessed on 8 February 2022). Polysomnographic recordings and annotations were downloaded from http://ceams-carsm.ca/en/mass/ (accessed on 20 April 2022) and https://github.com/klacourse/MODA_GC/tree/master/output/exp/annotFiles (accessed on 20 April 2022).

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
