# Peer review of "Two-Stage Atomic Decomposition of Multichannel EEG and the Previously Undetectable Sleep Spindles"

_sensors, 2024, doi:10.3390/s24030842_

Round 1

Reviewer 1 Report

Comments and Suggestions for Authors

The article titled "Two-stage atomic decomposition of multichannel EEG and the invisible sleep spindles" presents a two-step procedure for the atomic decomposition of multichannel EEG data. This approach, based on multivariate matching pursuit and dipolar inverse solution, aims to identify relevant EEG structures, with a particular focus on the often elusive sleep spindles.

One of the notable strengths of the article is the comprehensive parametrization of all detected EEG structures, including time and frequency centers, width (duration), phase, and spatial location of equivalent dipolar sources within the brain. This detailed analysis not only confirms the well-known prevalence of higher frequencies in posterior regions but also uncovers a significant gradient in the sagittal plane, providing valuable insights into the spatial distribution of sleep spindles.

The article is well-structured, with a clear presentation of the proposed methodology, however, I have some remarks and suggestions.

1. I encourage authors to contemplate expanding the existing literature review on spindles and EEG, along with their applications. Please refer to the following sources for further insights: https://doi.org/10.3390/e25091244 and https://doi.org/10.3390/brainsci13020275.

2. "From these decompositions, sleep spindles were selected as conforming to the criteria from Table 1, chosen to reflect the available prior knowledge"

Were the sleep spindles identified through manual or automated methods? If identified manually, please elaborate on the procedure used, including details such as whether a single person or a double examination was employed.

Finally, I encourage the author to delve into a discussion on the limitations of the presented work and explore potential applications, particularly in clinical contexts. This extension would provide a more comprehensive understanding of both the weaknesses and the practical implications of the study.

In conclusion, "Two-stage atomic decomposition of multichannel EEG and the invisible sleep spindles" is a well-executed and insightful study that brings attention to the limitations of existing methods in detecting sleep spindles. The proposed two-stage atomic decomposition approach, with its detailed parametrization and spatial analysis, opens new avenues for understanding the characteristics and distribution of sleep spindles in EEG data. 

Author Response

  1. I encourage authors to contemplate expanding the existing literature review on spindles and EEG, along with their applications. Please refer to the following sources for further insights: https://doi.org/10.3390/e25091244 and https://doi.org/10.3390/brainsci13020275.

We thank the Reviewer for the links to interesting papers related to the role of sleep spindles in research on Disorders of Consciousness—we will consider these references in our next study, which, accidentally, will address the application of proposed methodology in DoC. However, in the current paper we do not want to refer explicitly to DoC, as the method is far more general than even the sleep spindles as such, and we want to leave the Reader with the main impression of generality, while the spindles serve mostly as an example application.

  1. "From these decompositions, sleep spindles were selected as conforming to the criteria from Table 1, chosen to reflect the available prior knowledge"

Were the sleep spindles identified through manual or automated methods? If identified manually, please elaborate on the procedure used, including details such as whether a single person or a double examination was employed.

The whole procedure is fully automatic and repeatable, spindles are identified based upon parameters from Table 1 applied to the structures from a general MMP decomposition. We tried to highlight this fact in the revised version.

Finally, I encourage the author to delve into a discussion on the limitations of the presented work and explore potential applications, particularly in clinical contexts. This extension would provide a more comprehensive understanding of both the weaknesses and the practical implications of the study.

In the revised version we extended the discussion in the last section.

CHANGES IN THE RESUBMITTED MANUSCRIPT:

  • We added a flowchart of the procedure in a new Figure 1., and referred to it briefly in the text.
  • We added labeling of panels a, b, … in Figures 4, 5, 6 and 7, and modified the captions to refer to these labels.
  • We have moved the last paragraph of Section 3.3 to Discussion, renamed this section to “Discussion and conclusions”, and added one last paragraph.
  • We have changed the title to “Two-stage atomic decomposition of multichannel EEG and the previously undetectable sleep spindles”

Reviewer 2 Report

Comments and Suggestions for Authors

This article presents a method that combines the MMP algorithm with EEG source localization, and the methodology is validated using a dataset for sleep spindle detection. The authors of the MMP algorithm have published numerous articles and have developed a dedicated application software. However, before publication, I believe there are several aspects that need improvement in this article.

1、The title appears more like a review article. I suggest revising it to directly reflect the research content of this article.

2、For multiple figures, it is recommended to label them with sequential identifiers like a, b, c, etc. Can Figures 5 and 6 be merged?

3、I recommend including a survey of the integration of other time-frequency algorithms and source localization methods, such as wavelet analysis, in the introduction.

4、I suggest moving the data simulation from Section 3.3 to the discussion section. The original Section 4 (Discussion) appears more like a conclusion. Additionally, the content related to Figure 6 should still be retained in the discussion section.

5、I recommend adding an algorithmic flowchart in the methodology section. This addition would enhance the visual understanding of the model inputs and outputs discussed in Section 3.1.

6、I suggest providing a time-domain example from the MASS dataset for the SNR calculation in this paper. Low SNR commonly affects expert judgment, and the temporal waveform variations caused by EEG noise, artifacts, drift, etc., are entirely different. Which type of interference is the MP-DIP method proposed in this paper more suitable for?

7In the results of Section 3.1, what are the differences in the distribution of TP and FP between the SUMO, YASA methods, and MP-DIP? How does the comparison highlight the algorithmic superiority of MP-DIP in terms of interpretability?

8、Throughout the article, it's mentioned that the current method misses 2/3 of sleep spindle identifications. Can the MP-DIP method recognize these overlooked 2/3 of spindles? Does the detection of these 2/3 sleep spindles exceed the limits of EEG technique? In other words, is it challenging to detect and analyze these spindles using EEG methods, and is it necessary to resort to ECoG or other intracranial recording methods for detection?

9Currently, there are AIGC techniques using denoised diffusion probability models for EEG generation. In Section 3.3, what are the characteristics of the EEG data generated using the MVAR method in this article? Are there any illustrative diagrams or references to further explain this method?

Comments on the Quality of English Language

None

Author Response

1、The title appears more like a review article. I suggest revising it to directly reflect the research content of this article.

According to the suggestion, we have changed the title to “Two-stage atomic decomposition of multichannel EEG and the previously undetectable sleep spindles” which reflects exactly the two major components of the paper: methodology and example application.

2、For multiple figures, it is recommended to label them with sequential identifiers like a, b, c, etc. 

We added labeling of panels a, b, … in Figures 3, 4, 5 and 6 

Can Figures 5 and 6 be merged?

These figures illustrate two different effects: Fig. 5 shows the inaccuracies due to the MMP estimation of the parameters of detected graphoelements, while Fig. 6 simulates the situation in which MRI of the subject is not available to assess the errors of inverse solution, so in our opinion they should be separate.

3、I recommend including a survey of the integration of other time-frequency algorithms and source localization methods, such as wavelet analysis, in the introduction.

Introduction includes a concise survey of the existing combinations of time-frequency atomic decompositions and source localisation. Unfortunately, other time-frequency methods cannot be directly compared to atomic decompositions in relation to their efficiency as an input to the EEG/MEG source localisation, because only atomic decompositions return explicitly parameters of relevant structures, which can be automatically related to the parameters known from the tradition of visual EEG analysis. Similar use of other time-frequency estimators would require a complicated preprocessing before finding the parameters of relevant structures which could be used as an input to the inverse solutions. Quality of the overall results would depend dramatically on these choices (which are completely absent within the proposed procedure) and due to their potential variety the discussion would dramatically exceed the scope and expected volume of this manuscript.

4、I suggest moving the data simulation from Section 3.3 to the discussion section. The original Section 4 (Discussion) appears more like a conclusion. Additionally, the content related to Figure 6 should still be retained in the discussion section.

We thank the Reviewer for this remark, indeed, we did not notice this. Owing to this remark we changed the title of the last section to “Discussion and conclusions”, moved there the last paragraph of Section 3.3 and extended it slightly.

5、I recommend adding an algorithmic flowchart in the methodology section. This addition would enhance the visual understanding of the model inputs and outputs discussed in Section 3.1.

We added a flowchart in Figure 1.

6、I suggest providing a time-domain example from the MASS dataset for the SNR calculation in this paper. Low SNR commonly affects expert judgment, and the temporal waveform variations caused by EEG noise, artifacts, drift, etc., are entirely different. Which type of interference is the MP-DIP method proposed in this paper more suitable for?

The performance of the proposed method does not depend on the kind of noise obscuring the relevant structures (in this case sleep spindles) to the extent that even the background EEG activity is hereby treated as a noise. Representative examples of EEG can be found in one of several EEG Atlasses, while the calculation of SNR is explicitly described in Eq. (5) 

7、In the results of Section 3.1, what are the differences in the distribution of TP and FP between the SUMO, YASA methods, and MP-DIP? How does the comparison highlight the algorithmic superiority of MP-DIP in terms of interpretability?

Comparisons to SUMO and YASA cannot highlight the superiority of MP-dip in terms of interpretability, because, contrary to MP-dip, these methods do not provide an explicit parameterization of the detected structures. As for the concordance with visual detection, MP-dip is not optimized for its maximization, so we do not expect a superiority in this aspect, just a general concordance.

8、Throughout the article, it's mentioned that the current method misses 2/3 of sleep spindle identifications. Can the MP-DIP method recognize these overlooked 2/3 of spindles? Does the detection of these 2/3 sleep spindles exceed the limits of EEG technique? In other words, is it challenging to detect and analyze these spindles using EEG methods, and is it necessary to resort to ECoG or other intracranial recording methods for detection?

Yes, these overlooked ⅔ of spindles are automatically detected by the proposed approach, and in this manuscript they are termed “false positives” in relation to human detections. They are included in the exemplary computations of the frequency gradients in Fig. 4. Detection of these spindles does not require any extra measurements, the proposed method can find these structures in the EEG time series in spite of unfavourable SNR, which obscured them to human scorers and methods designed to replicate human detections. 

9、Currently, there are AIGC techniques using denoised diffusion probability models for EEG generation. In Section 3.3, what are the characteristics of the EEG data generated using the MVAR method in this article? Are there any illustrative diagrams or references to further explain this method?

For the estimation of the accuracy of parameters estimation we needed a model providing explicitly the ground truth values for parameters, hence we chose a simple linear model where the background EEG, generated from the multivariate AR model, has the statistical properties matching the analyzed EEG, and sleep spindles are simulated as Gabor functions linearly added to the simulated background EEG with amplitudes matching the forward solution of an equivalent dipolar source. This allows us to investigate only the performance of the signal processing fit, without the influence of more complicated hypotheses of EEG generation.

CHANGES IN THE RESUBMITTED MANUSCRIPT:

  • We added a flowchart of the procedure in a new Figure 1., and referred to it briefly in the text.
  • We added labeling of panels a, b, … in Figures 4, 5, 6 and 7, and modified the captions to refer to these labels.
  • We moved the last paragraph of Section 3.3 to Discussion, renamed this section to “Discussion and conclusions”, and added one last paragraph.
  • We have changed the title to “Two-stage atomic decomposition of multichannel EEG and the previously undetectable sleep spindles”

Round 2

Reviewer 1 Report

Comments and Suggestions for Authors

The Authors have addressed all of my concerns with the original manuscript. The revised manuscript is ready for publication